# Cross-sectional analyses of participation in cancer screening and use of hormone replacement therapy and medications in meat eaters and vegetarians: the EPIC-Oxford study

Tammy Y N Tong, Paul N Appleby, Kathryn E Bradbury, Timothy J Key

Cancer Epidemiology Unit, Nuffield Department of Population Health, University of Oxford, Oxford, UK

**Correspondence to**
Dr Tammy Y N Tong;
tammy.tong@ndph.ox.ac.uk

## ABSTRACT

**Objectives** To examine differences in health-related behaviours such as screening or testing for cancer, use of hormone replacement therapy (HRT) and use of other medications in different diet groups.

**Design** We studied 31 260 participants across four diet groups (18 155 meat eaters, 5012 fish eaters, 7179 vegetarians, 914 vegans) in the UK EPIC-Oxford cohort. Information was collected in 5-year (around 2000–2003) or 10-year (around 2007) follow-up questionnaires regarding participation in breast screening, cervical screening, prostate-specific antigen (PSA) testing, use of HRT and use of medications for the past 4 weeks. Using Poisson regression, we estimated the prevalence ratios (PR) for each behaviour across people of different diet groups, using meat eaters as the reference group.

**Results** Compared with meat eaters, vegetarian (PR: 0.94, 95% CI 0.89 to 0.98) and vegan (PR: 0.82, 95% CI 0.71 to 0.95) women reported lower participation in breast screening, and vegetarian men were less likely to report PSA testing (PR: 0.82, 95% CI 0.71 to 0.96). No differences were observed among women for cervical screening. In women, all non-meat-eating groups reported lower use of HRT compared with meat eaters (P heterogeneity <0.0001). Lower reported use of any medication was observed for participants in all non-meat-eating groups with no (P<0.0001) or one (P=0.0002) self-reported illness. No heterogeneity was observed across the diet groups for the reported use of specific medication for high blood pressure, high blood cholesterol, asthma, diabetes and thyroid disease.

**Conclusions** Differences in self-reported breast screening, PSA testing, HRT use and overall medication use were observed across the diet groups. Whether such differences contribute to differential long-term disease risks requires further study.

## INTRODUCTION

People of different habitual diet groups have been shown to have different health characteristics. Compared with meat eaters, vegetarians generally have lower body mass index, blood pressure and circulating low-density

### Strengths and limitations of this study

► This study is the first to simultaneously examine the reported uptake of breast and cervical cancer screening, prostate-specific antigen testing, hormone replacement therapy (HRT) use and medication use in different diet groups.
► The study includes a large number of participants recruited from across different regions in the UK, with a high proportion of fish eaters, vegetarians and vegans.
► Recall bias is possible because assessment of cancer screening or testing, HRT use and medication use was based on self-report, although there is no indication that such misclassification bias should differ by diet group.
► The study is cross-sectional and we cannot infer causality.

lipoprotein cholesterol levels,[1–3] characteristics likely to reduce disease risk. However, evidence on the long-term risk of many non-communicable diseases across people of different diet groups is limited.

For cancer risk, both a UK[4] and a US[5] study reported lower risk of overall cancer incidence with a vegetarian diet. Because health-related behaviours, such as participation in cancer screening[6] or use of hormone replacement therapy (HRT),[7 8] may contribute to the observed rates of cancer, the presence of any differences in these behaviours between diet groups in different populations deserves further investigation. Results from a Swedish cohort[9] and a US cohort[10] showed that vegetarians (including vegans and people who ate fish but not meat) had lower odds of attending breast screening and prostate cancer screening, respectively, when compared with meat eaters, and vegetarians also had lower use of HRT compared with non-vegetarians.[5]

For cardiovascular diseases, vegetarians in EPIC-Oxford have been observed to have lower ischaemic heart disease risk (hospitalisation and death combined),[11] but no significant difference in ischaemic heart disease mortality was observed between diet groups in the same population.[12] The reason for this apparent difference between incidence and mortality is unclear. One possible explanation could be the differential use of appropriate medications in the different diet groups, which subsequently influences disease mortality. In a Belgian population, for example, vegetarians had lower use of prescription medications compared with non-vegetarians, but similar use of non-prescription drugs.[13]

The increasing popularity and interest in vegetarian diets[14] prompts research on the long-term health of vegetarians and vegans. Because health behaviours such as screening or medication use may ultimately influence disease risk, the understanding of any differences in these behaviours by diet group is crucial for the appropriate appraisal of possible differences in disease risk between diet groups. However, current knowledge on this topic is insufficient because literature on participation in screening and use of medication across people of different diet groups is scarce. Therefore, the aim of this study was to assess some of these relevant health behaviours, including participation in cancer screening or testing, and use of HRT and other medications among people of different diet groups, in a large population-based cohort in the UK with a high percentage of vegetarians.

## METHODS
### Study population
The EPIC-Oxford study is a UK-based cohort recruited between 1993 and 1999. Participants gave written informed consent. Details of the recruitment process have been described previously.[1] In brief, a combination of general practitioner (GP) recruitment and postal recruitment was used. The GP recruitment invited men and women aged 35–59 years registered with participating GPs and recruited 7421 participants. The postal recruitment was targeted at vegetarians, vegans and other people interested in diet and health, by contacting members of the Vegetarian Society and The Vegan Society, and via leaflets enclosed in vegetarian and health food magazines and displayed in health-food shops, and recruited 57 990 participants aged ≥20 years. Altogether, 57 443 participants completed a full recruitment questionnaire, which asked about their personal details (including postcode to which a Townsend index of area-level deprivation was assigned),[15] habitual diet, and other health and lifestyle characteristics, including personal and family medical history, medication use, socioeconomic characteristics, smoking and drinking behaviours, and physical activity levels. A follow-up questionnaire was sent to surviving participants approximately 5 years after recruitment (mostly from 2000 to 2003), and a second follow-up questionnaire was mailed approximately 10 years after recruitment (mostly in 2007). In the follow-up questionnaires, updated information was gathered on diet, health and lifestyle, including self-reported current health. Due to the changing research focus over the course of data collection, slight variations existed between questions asked on the 5-year and 10-year follow-up questionnaires.

### Assessment of diet group
In the recruitment questionnaire and each subsequent follow-up questionnaire, four questions were asked regarding consumption of meat, fish, dairy products and eggs, in the form of 'Do you eat any meat?' or similar for the other three food groups. Responses to these questions were used to assign participants to one of four diet groups at each time point: meat eaters (participants who ate meat, irrespective of whether they ate fish, dairy products or eggs), fish eaters (participants who did not eat meat but did eat fish), vegetarians (participants who did not eat meat or fish, but did eat one or both of dairy products and eggs) and vegans (participants who did not eat meat, fish, dairy products or eggs).

### Assessment of participation in screening, HRT and medication use
In the follow-up questionnaires, women were asked if they had ever had a breast screening by mammography, cervical screening by smear test (only on the 5-year follow-up questionnaire) or used HRT, and men were asked if they had ever had a prostate-specific antigen (PSA) test (only on the 10-year follow-up questionnaire). On the 10-year follow-up questionnaire, all participants were asked if they had used any medication for most of the last 4 weeks, with 36 named medications and a free text field for reporting regular use of any medication not on the list; participants were also asked if they had been diagnosed with any of a list of 29 medical conditions, and the year when the condition was first diagnosed. The full list of the 36 medications and 29 medical conditions is given in online supplementary text 1 and 2. The corresponding question on medication use on the 5-year questionnaire was shorter, with 20 named medications and 26 medical conditions.

For assessment of specific medication use, five common medical conditions associated with specific medications were identified: high blood pressure (commonly treated with one or more of amlodipine, enalapril, frusemide, propranolol, atenolol, bendrofluazide, lisinopril and nifedipine), high blood cholesterol (atorvastatin and simvastatin), asthma (beclomethasone and salbutamol), diabetes (insulin and metformin) and thyroid disease (thyroxine).

### Statistical analyses
Information on assignment to diet group and assessment of health behaviour from the 10-year follow-up questionnaire was used for our analyses, except for the assessment of participation in cervical screening, which was only asked on the 5-year follow-up questionnaire. Participants

**Table 1** Characteristics by diet group of participants in the EPIC-Oxford study who completed the second follow-up questionnaire (n=31 260)*

| Characteristics | Meat eaters | Fish eaters | Vegetarians | Vegans | Total |
|---|---|---|---|---|---|
| Number of participants (% female) | 18 155 (78.2) | 5012 (81.8) | 7179 (76.3) | 914 (66.1) | 31 260 (78.0) |
| Mean (SD) age at questionnaire completion, years | 58.9 (12.5) | 53.8 (12.5) | 51.6 (12.7) | 50.7 (12.3) | 56.1 (13.0) |
| Smoking status†, n (%) | | | | | |
| Never smoker | 10 073 (55.7) | 2786 (55.6) | 4339 (60.5) | 547 (59.9) | 17 745 (56.9) |
| Former smoker | 6927 (38.3) | 1961 (39.2) | 2460 (34.3) | 330 (36.1) | 11 678 (37.5) |
| Current smoker | 1094 (6.0) | 260 (5.2) | 367 (5.1) | 36 (3.9) | 1757 (5.6) |
| Mean (SD) alcohol consumption, g/day | 8.7 (9.3) | 8.2 (8.7) | 7.6 (8.9) | 6.7 (9.2) | 8.3 (9.1) |
| Self-reported current health†, n (%) | | | | | |
| Excellent | 3713 (21.9) | 1323 (28.1) | 1950 (28.7) | 325 (37.2) | 7311 (24.9) |
| Good | 9962 (58.8) | 2688 (57.0) | 3851 (56.6) | 446 (51.0) | 16 947 (57.8) |
| Fair | 2858 (16.9) | 612 (13.0) | 876 (12.9) | 80 (9.2) | 4426 (15.1) |
| Poor | 400 (2.4) | 92 (2.0) | 122 (1.8) | 23 (2.6) | 637 (2.2) |
| Townsend Deprivation Index†, n (%) | | | | | |
| Richest category | 4463 (27.6) | 984 (21.8) | 1542 (23.7) | 153 (18.3) | 7141 (25.5) |
| Poorest category | 3438 (21.2) | 1207 (26.8) | 1732 (26.7) | 285 (34.1) | 6662 (23.8) |
| In same diet group at recruitment, n (%) | 15 908 (87.7) | 3057 (61.1) | 6373 (89.1) | 573 (62.7) | 25 911 (83.0) |
| Taking medication in the past 4 weeks, n (%) | 10 196 (56.2) | 2105 (42.0) | 2829 (39.4) | 255 (27.9) | 15 385 (49.2) |
| Number of reported illnesses and conditions, n (%) | | | | | |
| None | 4455 (24.5) | 1635 (32.6) | 2603 (36.3) | 344 (37.6) | 9037 (28.9) |
| One | 4724 (26.0) | 1472 (29.4) | 2170 (30.2) | 291 (31.8) | 8657 (27.7) |
| Two | 3682 (20.3) | 906 (18.1) | 1261 (17.6) | 154 (16.8) | 6003 (19.2) |
| Three | 2404 (13.2) | 524 (10.5) | 630 (8.8) | 74 (8.1) | 3632 (11.6) |
| Four or more | 2890 (15.9) | 475 (9.5) | 515 (7.2) | 51 (5.6) | 3931 (12.6) |
| Reported high blood pressure†, n (%) | 4397 (29.2) | 686 (16.2) | 944 (15.2) | 85 (10.6) | 6112 (23.2) |
| and taking appropriate medication, n (%) | 2573 (58.5) | 357 (52.0) | 430 (45.6) | 40 (47.1) | 3400 (55.6) |
| Reported high blood cholesterol†, n (%) | 3351 (23.1) | 561 (13.5) | 645 (10.5) | 44 (5.5) | 4601 (18.0) |
| and taking appropriate medication, n (%) | 1646 (49.1) | 209 (37.3) | 243 (37.7) | 14 (31.8) | 2112 (45.9) |
| Reported asthma†, n (%) | 1885 (13.6) | 496 (12.1) | 758 (12.4) | 88 (11.1) | 3227 (12.9) |
| and taking appropriate medication, n (%) | 737 (39.1) | 169 (34.1) | 246 (32.5) | 17 (19.3) | 1169 (36.2) |
| Reported diabetes†, n (%) | 707 (5.2) | 75 (1.9) | 119 (2.0) | 7 (0.9) | 908 (3.7) |
| and taking appropriate medication, n (%) | 446 (63.1) | 41 (54.7) | 84 (70.6) | 6 (85.7) | 577 (63.5) |
| Reported thyroid disease†, n (%) | 1545 (11.1) | 380 (9.2) | 465 (7.6) | 56 (7.1) | 2446 (9.8) |
| and taking appropriate medication, n (%) | 1191 (77.1) | 273 (71.8) | 337 (72.5) | 37 (66.1) | 1838 (75.1) |

*Based on participant characteristics at the time of the second follow-up questionnaire (completed approximately 10 years from baseline, around 2007).
†Unknown for some participants.

were excluded from all analyses if they did not answer the relevant questions to be assigned to an appropriate diet group (n=28). In order to ensure that an overlapping population was used for the analyses of all outcomes, participants were also excluded if they did not answer the relevant question on medication use (n=407). For the analyses related to participation in breast screening, cervical screening, PSA testing or HRT use, only women or men who answered the relevant question and were in the specified age group at questionnaire completion were included. The age group specifications were as follows: age 50–74 years for breast screening, age 25–74 years for cervical screening, age 50–84 years for PSA testing and age 50–74 years for HRT use. For HRT use, we further restricted the analysis to postmenopausal women, determined by including only participants who answered yes to the question 'Have you been through your menopause?' on the follow-up questionnaire.

**Table 2** Participation in cancer screening or testing by diet group of women and men in the EPIC-Oxford study

| Cancer screening or testing/diet group | Participants answering the relevant question, n* | Participants answering in the affirmative, n (%)* | Prevalence ratio (95% CI)* |
|---|---|---|---|
| Breast screening† | | | |
| Meat eaters | 9239 | 8813 (95.4) | 1.00 (ref) |
| Fish eaters | 2143 | 1928 (90.0) | 0.96 (0.92 to 1.01) |
| Vegetarians | 2395 | 2078 (86.8) | 0.94 (0.89 to 0.98) |
| Vegans | 239 | 182 (76.2) | 0.82 (0.71 to 0.95) |
| | | | P heterogeneity=0.004 |
| Cervical screening‡ | | | |
| Meat eaters | 15936 | 15365 (96.4) | 1.00 (ref) |
| Fish eaters | 4513 | 4369 (96.8) | 1.00 (0.97 to 1.03) |
| Vegetarians | 6574 | 6268 (95.3) | 0.98 (0.95 to 1.01) |
| Vegans | 758 | 691 (91.2) | 0.94 (0.87 to 1.02) |
| | | | P heterogeneity=0.37 |
| Prostate-specific antigen testing§ | | | |
| Meat eaters | 3078 | 1066 (34.6) | 1.00 (ref) |
| Fish eaters | 594 | 181 (30.5) | 0.99 (0.85 to 1.17) |
| Vegetarians | 947 | 228 (24.1) | 0.82 (0.71 to 0.96) |
| Vegans | 164 | 33 (20.1) | 0.72 (0.50 to 1.02) |
| | | | P heterogeneity=0.023 |

*Number answering the relevant question and n (%) answering in the affirmative were as observed. Prevalence ratios were adjusted for age at follow-up (<40, 40–44, 45–49, 50–54, 55–59, 60–64, 65–69, 70–74, ≥75 years, as appropriate according to the age range of included participants), region of residence (eight regions) and self-reported current health (excellent, good, fair, poor, unknown).
†Included women aged 50–74 who answered the relevant question on the second (10-year) follow-up questionnaire.
‡Included women aged 25–74 who answered the relevant question on the first (5-year) follow-up questionnaire.
§Included men aged 50–84 who answered the relevant question on the second (10-year) follow-up questionnaire.

For each analysis, we used Poisson regression to estimate prevalence ratios (95% CI) of cancer screening or testing (breast screening, cervical screening, PSA testing), HRT use or medication use in different diet groups, using meat eaters as the reference group. For analyses of cancer screening or testing and use of HRT, we adjusted for age at follow-up (<40, 40–44, 45–49, 50–54, 55–59, 60–64, 65–69, 70–74, ≥75 years as appropriate for the age range included in the analysis), region of recruitment (eight geographical regions across the UK) and self-reported current health (excellent, good, fair, poor, unknown). For analyses of any medication use, we adjusted for the cross-stratification of sex and age at follow-up, region of recruitment, self-reported current health, and the number of self-reported illnesses or conditions (0, 1, 2, 3, ≥4). To further assess whether any variation in medication use by diet group varied by health status, we repeated the analyses stratified by the number of self-reported illnesses or conditions using the above categorisation. Subsequently, for each of high blood pressure, high blood cholesterol, asthma, diabetes and thyroid disease, we estimated the prevalence ratios of taking appropriate medication by diet group among people diagnosed with each condition, in turn adjusting for covariates as above and additionally for years since reported diagnosis, calculated as year of

follow-up questionnaire completion minus reported year of diagnosis (<2, 2–3, 4–5, 6–9, ≥10 years, unknown).

As sensitivity analyses, we repeated the analyses as follows: using data from the 5-year follow-up questionnaire where available, and further adjusting for smoking status (never, former, current, unknown), alcohol consumption (<1 g/day, 1–7 g/day, 8–15 g/day, ≥16 g/day), Townsend index of area-level deprivation (quartiles and unknown) and education level (no qualifications; basic secondary, eg, O level; higher secondary, eg, A level; degree, unknown). All statistical analyses were performed using Stata V.14.1 (Stata Corp, Texas, United States), and P values <0.05 were considered statistically significant.

## RESULTS
### Cohort characteristics
Overall, 57443 participants in EPIC-Oxford cohort completed a full recruitment questionnaire, of whom 38043 (66%) completed the 5-year follow-up questionnaire and 31695 (55%) completed the 10-year follow-up questionnaire. After excluding participants who did not answer the relevant questions on diet group or on medication use, data for 31260 participants who completed the 10-year follow-up questionnaire (18155 meat eaters,

**Table 3** Use of hormone replacement therapy by diet group of women in the EPIC-Oxford study

| Diet group | Participants answering the relevant question, n* | Participants answering in the affirmative, n (%)* | Prevalence ratio (95% CI)* |
|---|---|---|---|
| Meat eaters | 6911 | 3098 (44.8) | 1.00 (ref) |
| Fish eaters | 1614 | 541 (33.5) | 0.80 (0.73 to 0.88) |
| Vegetarians | 1778 | 541 (30.4) | 0.74 (0.68 to 0.81) |
| Vegans | 188 | 31 (16.5) | 0.42 (0.30 to 0.60) |
| | | | P heterogeneity <0.0001 |

*Number answering the relevant question and n (%) answering in the affirmative were as observed. Prevalence ratios were adjusted for age at follow-up (50–54, 55–59, 60–64, 65–69, 70–74 years), region of residence (eight regions) and self-reported current health (excellent, good, fair, poor, unknown). Included postmenopausal women aged 50–74 who answered the relevant question on the second (10-year) follow-up questionnaire.

5012 fish eaters, 7179 vegetarians and 914 vegans) were used for most of the analyses. Characteristics of the participants are presented in table 1. Overall, non-meat eaters were younger, more likely to report having excellent health, less likely to report taking medication in the past 4 weeks and less likely to have reported any illnesses or conditions.

### Participation in screening and use of HRT and medications
Overall, 14 016 women were included in the analyses for breast screening, 27 781 women for cervical screening and 4783 men for PSA testing (table 2). In women, compared with meat eaters, vegetarians (prevalence ratio: 0.94, 95% CI 0.89 to 0.98) and vegans (prevalence ratio: 0.82, 95% CI 0.71 to 0.95), but not fish eaters (prevalence ratio: 0.96, 95% CI 0.92 to 1.01), had lower reported attendance of breast screening, but no significant heterogeneity was observed between the diet groups for reported participation in cervical screening (P heterogeneity=0.37). In men, vegetarians had lower reported uptake of PSA testing (prevalence ratio: 0.82, 95% CI 0.71 to 0.96) than meat eaters, while the difference in uptake appeared lower but did not reach statistical significance in vegans (prevalence ratio: 0.72, 95% CI 0.50 to 1.02) and was not significantly different in fish eaters (prevalence ratio: 0.99, 95% CI 0.85 to 1.07). For HRT use, women who were non-meat eaters reported lower use (fish eaters—prevalence ratio: 0.80, 95% CI 0.73 to 0.88; vegetarians—prevalence ratio: 0.74, 95% CI 0.68 to 0.81; vegans—prevalence ratio: 0.42, 95% CI 0.30 to 0.60) compared with women who were meat eaters (table 3).

Irrespective of the number of self-reported illnesses and conditions, non-meat eaters reported lower use of any medication (fish eaters—prevalence ratio: 0.92, 95% CI 0.87 to 0.96; vegetarians—prevalence ratio 0.93, 95% CI 0.89 to 0.98; vegans—prevalence ratio: 0.71, 95% CI 0.63 to 0.81) compared with meat eaters (table 4). When the analyses were stratified by the number of self-reported illnesses or conditions, non-meat eaters with no (P<0.0001) or one (P=0.0002) illness or condition reported lower medication use compared with meat eaters, but the association was attenuated and no longer statistically significant among participants with two, three,

or four or more illnesses or conditions. For medication use specific to several common illnesses and conditions, no significant differences were observed between the diet groups in the reported use of appropriate medications for high blood pressure, high blood cholesterol, asthma, diabetes or thyroid disease, among participants diagnosed with each of these conditions (table 5). Results were consistent when we repeated the analyses where possible using data from the 5-year follow-up questionnaire, or when we further adjusted for smoking, alcohol consumption, Townsend Deprivation Index and education level (online supplementary table 1).

### DISCUSSION
### Summary of results
In this UK population-based cohort with a large proportion of participants from different diet groups, we generally observed lower participation in breast screening and lower HRT use among women who were non-meat eaters (separately categorised as fish eaters, vegetarians and vegans) compared with women who were meat eaters. Vegetarian men had lower participation in PSA testing compared with meat eating men, but no significant difference was observed for cervical screening in women across the diet groups. For medication use, non-meat eaters were less likely to report taking medications than meat eaters overall, but there were no significant differences in medication use among people reporting two or more illnesses or conditions, or for people reporting taking specific medications for various self-reported conditions.

### Comparison with other studies
Few studies have reported on the participation in cancer screening or testing, HRT use or medication use among people of different diet groups, and no study has assessed all these behaviours simultaneously in the same cohort. For breast cancer screening, consistent with our findings, the Swedish Malmö Diet and Cancer Study reported that non-attendance for breast cancer screening was more likely in people who were vegetarians or vegans (OR: 1.49, 95% CI 1.11 to 1.99).[9] Analyses of data from the Adventist Health Study-2 in the USA and

**Table 4** Medication use by number of self-reported illnesses or conditions and diet group of participants in the EPIC-Oxford study*

| Number of self-reported illnesses or conditions/diet group | Participants, n† | Percentage taking any medication† | Prevalence ratio (95% CI)† |
|---|---|---|---|
| Any number‡ | | | |
| Meat eaters | 18155 | 56.2 | 1.00 (ref) |
| Fish eaters | 5012 | 42.0 | 0.92 (0.87 to 0.96) |
| Vegetarians | 7179 | 39.4 | 0.93 (0.89 to 0.98) |
| Vegans | 914 | 27.9 | 0.71 (0.63 to 0.81) |
| | | | P heterogeneity <0.0001 |
| None | | | |
| Meat eaters | 4455 | 16.9 | 1.00 (ref) |
| Fish eaters | 1635 | 11.9 | 0.80 (0.68 to 0.94) |
| Vegetarians | 2603 | 11.5 | 0.80 (0.70 to 0.92) |
| Vegans | 344 | 6.1 | 0.47 (0.30 to 0.72) |
| | | | P heterogeneity <0.0001 |
| One | | | |
| Meat eaters | 4724 | 48.9 | 1.00 (ref) |
| Fish eaters | 1472 | 39.1 | 0.87 (0.80 to 0.96) |
| Vegetarians | 2170 | 40.5 | 0.91 (0.84 to 0.99) |
| Vegans | 291 | 29.2 | 0.69 (0.55 to 0.85) |
| | | | P heterogeneity=0.0002 |
| Two | | | |
| Meat eaters | 3682 | 66.9 | 1.00 (ref) |
| Fish eaters | 906 | 58.8 | 0.94 (0.86 to 1.04) |
| Vegetarians | 1261 | 58.1 | 0.97 (0.89 to 1.06) |
| Vegans | 154 | 42.2 | 0.74 (0.58 to 0.95) |
| | | | P heterogeneity=0.082 |
| Three | | | |
| Meat eaters | 2404 | 82.6 | 1.00 (ref) |
| Fish eaters | 524 | 74.0 | 0.94 (0.84 to 1.05) |
| Vegetarians | 630 | 73.0 | 0.94 (0.84 to 1.04) |
| Vegans | 74 | 59.5 | 0.78 (0.57 to 1.05) |
| | | | P heterogeneity=0.22 |
| Four or more | | | |
| Meat eaters | 2890 | 93.0 | 1.00 (ref) |
| Fish eaters | 475 | 86.9 | 0.96 (0.86 to 1.06) |
| Vegetarians | 515 | 88.9 | 0.98 (0.89 to 1.09) |
| Vegans | 51 | 78.4 | 0.87 (0.63 to 1.19) |
| | | | P heterogeneity=0.70 |

*Refers to medication use for most of the past 4 weeks on the second (10-year) follow-up questionnaire, excluding hormone replacement therapy and contraceptive pills.
†Number of participants and percentage taking any medication were as observed. Prevalence ratios were adjusted for the cross-classification of sex and age at follow-up (<40, 40–44, 45–49, 50–54, 55–59, 60–64, 65–69, 70–74, ≥75 years), region of residence (eight regions) and self-reported current health (excellent, good, fair, poor, unknown).
‡Prevalence ratios for this category were further adjusted for the number of self-reported illnesses or conditions (0, 1, 2, 3, ≥4).

Canada showed that all non-meat eaters were less likely to report PSA testing compared with meat eaters (OR: 0.79, 95% CI 0.66 to 0.95 for fish eaters; OR: 0.76, 95% CI 0.67 to 0.86 for vegetarians; and OR: 0.50, 95% CI 0.42 to 0.60 for vegans),[10] whereas we only observed a lower reported uptake among the vegetarians but not the

**Table 5** Medication use for specific conditions by diet group of participants in the EPIC-Oxford study*

| Condition/diet group | Participants reporting the condition, n (mean years since reported diagnosis)† | Participants taking appropriate medication, n (%)† | Prevalence ratio (95% CI)† |
|---|---|---|---|
| High blood pressure‡ | | | |
| Meat eaters | 4397 (9.8) | 2573 (58.5) | 1.00 (ref) |
| Fish eaters | 686 (9.3) | 357 (52.0) | 0.97 (0.86 to 1.08) |
| Vegetarians | 944 (9.0) | 430 (45.6) | 0.91 (0.82 to 1.01) |
| Vegans | 85 (9.0) | 40 (47.1) | 0.92 (0.67 to 1.26) |
| | | | P heterogeneity=0.37 |
| High blood cholesterol§ | | | |
| Meat eaters | 3351 (6.3) | 1646 (49.1) | 1.00 (ref) |
| Fish eaters | 561 (5.3) | 209 (37.3) | 0.88 (0.76 to 1.01) |
| Vegetarians | 645 (5.5) | 243 (37.7) | 0.94 (0.81 to 1.08) |
| Vegans | 44 (7.1) | 14 (31.8) | 0.74 (0.44 to 1.26) |
| | | | P heterogeneity=0.20 |
| Asthma¶ | | | |
| Meat eaters | 1885 (25.3) | 737 (39.1) | 1.00 (ref) |
| Fish eaters | 496 (23.2) | 169 (34.1) | 0.98 (0.82 to 1.17) |
| Vegetarians | 758 (23.4) | 246 (32.5) | 0.97 (0.84 to 1.14) |
| Vegans | 88 (27.9) | 17 (19.3) | 0.67 (0.41 to 1.09) |
| | | | P heterogeneity=0.45 |
| Diabetes** | | | |
| Meat eaters | 707 (10.0) | 446 (63.1) | 1.00 (ref) |
| Fish eaters | 75 (14.8) | 41 (54.7) | 0.78 (0.56 to 1.08) |
| Vegetarians | 119 (10.6) | 84 (70.6) | 1.05 (0.81 to 1.35) |
| Vegans | 7 (13.2) | 6 (85.7) | 1.07 (0.45 to 2.51) |
| | | | P heterogeneity=0.46 |
| Thyroid disease†† | | | |
| Meat eaters | 1545 (13.2) | 1191 (77.1) | 1.00 (ref) |
| Fish eaters | 380 (11.6) | 273 (71.8) | 0.95 (0.83 to 1.09) |
| Vegetarians | 465 (11.2) | 337 (72.5) | 0.97 (0.85 to 1.10) |
| Vegans | 56 (11.8) | 37 (66.1) | 0.88 (0.63 to 1.22) |
| | | | P heterogeneity=0.78 |

*Refers to medication use for most of the past 4 weeks specific to the condition described among participants who reported diagnosis for the condition on the second (10-year) follow-up questionnaire.

†Number reporting the condition (mean years since reported diagnosis) and number (%) taking appropriate medication were as observed. Prevalence ratios were adjusted for the cross-classification of sex and age at follow-up (<40, 40–44, 45–49, 50–54, 55–59, 60–64, 65–69, 70–74, ≥75 years), region of residence (eight regions), self-reported current health (excellent, good, fair, poor, unknown), years since reported diagnosis (calculated as year of follow-up questionnaire completion minus reported year of diagnosis; <2, 2–3, 4–5, 6–9, ≥10 years, unknown), and number of self-reported illnesses or conditions (1, 2, 3, ≥4).

‡Reported use of at least one of amlodipine, enalapril, frusemide, propranolol, atenolol, bendrofluazide, lisinopril and nifedipine.

§Reported use of at least one of atorvastatin and simvastatin.

¶Reported use of at least one of beclomethasone and salbutamol.

**Reported use of at least one of insulin and metformin.

††Reported use of thyroxine.

fish eaters (nor the vegans, perhaps because of limited numbers) compared with meat eaters in EPIC-Oxford. However, given the much higher rates of PSA testing in the Adventist Health Study-2 (73.3% vs 31.5% in EPIC-Oxford), attitudes towards screening are likely to be different in the two populations, and therefore the results might not be directly comparable. Similar to our study, the Adventist Health Study-2 also reported lower ever use of HRT (adjusted for age and race) in pescovegetarians (21.0%) and lactovegetarians (20.4%), and the

lowest use in vegans (16.2%), when compared with non-vegetarians (22.4%).[5]

For medication use, a cross-sectional study in a Belgian population reported lower use of prescribed medications when comparing vegetarians with a reference Belgian population (25.5% vs 47.3%, P<0.001).[13] While this is consistent with our findings on overall medication use, the study did not assess the use of medications stratified by the number of illnesses, nor did they assess appropriate medication use for specific medical conditions. No studies were found which examined participation of cervical screening among people of different diet groups.

### Interpretation of findings and implications

Our findings indicate differences in some health-related behaviours between people of different diet groups, although the reasons behind such differences are unclear. For the observed differences in screening rates, possible explanations could be related to different attitudes towards the screening programmes. In the UK since 1988,[6 16] all women aged 50–70 are invited to attend breast cancer screening clinics,[17] and all women aged 25–64 are invited for cervical screening[18] at regular intervals. On the other hand, there is no national programme for PSA testing, although men over the age of 50 are eligible to arrange for testing via their GP if they wish.[19] In studies that assessed attitudes towards cancer screening or testing, common reasons that affect people's participation in screening include their education level and knowledge of the procedure, recommendation by their doctor, fear of the procedure or the outcome, or their perceived risk of cancer.[20–23] If vegetarians and vegans felt their diets or lifestyles were protective against cancer for example, they might be more likely to forgo cancer screening as a result of lower perceived risk. However, no information was found on whether or how such attitudes may vary by diet group.

In a small focus group study in Scotland which asked participants about their attitudes towards cancer screening (n=31 for cervical screening, n=10 for breast screening), the study participants reported that they felt pressure from healthcare professionals, family and friends to attend cervical screening but not breast screening, and that they also considered cervical screening to be normative routine behaviour.[24] Such differences in attitudes towards breast screening and cervical screening are of interest, as these may help to explain the differences we observed in participation for breast screening but not cervical screening, if the latter was considered routine behaviour. However, relevant evidence is lacking, and both dietary and non-dietary factors that are associated with attendance for either breast screening or PSA testing deserve further study.

Reasons for the observed lower prevalence of HRT use and medication use among people of different diet groups are also unclear. The prevalence of medication use in meat eaters (56%) in EPIC-Oxford was slightly higher than the UK average of 43% of men and 50% of women aged 16 or above who reported taking at least one prescribed medicine in the last week,[25] confirming the relatively low prevalence of medication use in the vegetarians (39%) and vegans (28%). However, given the differences in age ranges and possible differences in medications accounted for, strict comparisons cannot be made. Because lower reported use of medications was observed even in people with no (especially) or only one reported illness or condition, better health among non-meat eaters is unlikely to be the only or a sufficient explanation for the differences. Non-meat eaters may also be reluctant to take medications that are likely to contain animal-derived products,[26] or may prefer to use homoeopathic medications[13] or other alternative therapies. Since information on medication use in this study was based on a prespecified list from the follow-up questionnaire, it was not possible to assess the use of alternative therapies or any other named medications, despite their possible contributions to prevalence of overall medication use.

Differential participation in screening for breast or prostate cancer, use of HRT and use of medications for people of distinct diet groups may ultimately lead to differences in disease incidence or prognosis due to possible detection bias and differential postdiagnosis treatment. For example, breast cancer screening results in higher incidence but reduced mortality from breast cancer among those who are screened.[6] Prostate cancer testing is also linked to increased incidence in those who are tested.[27 28] Therefore, using breast cancer as an example, given the lower rates of breast cancer screening among non-meat-eating women both in EPIC-Oxford and in the Swedish Malmö Diet and Cancer Study,[9] it is possible that the observed incidence of breast cancer in these diet groups underestimates the true incidence owing to detection bias, but that ultimately these women would be expected to have a somewhat higher mortality from breast cancer. Therefore, future work on assessing breast cancer risk in people of different diet groups should take into account any differences in screening rates between diet groups.

Similarly, it is not clear why there was differential use of HRT in the four diet groups, for example whether it was because non-meat eaters were less likely to have symptoms or because they were less likely to seek treatment when symptoms appear. Regardless of the underlying reason, the observed lower reported use of HRT among non-meat-eating women deserves attention, because use of HRT may confound any observed associations between diet group and breast cancer, given that HRT preparations containing oestrogens and progestogens have been shown to increase the risk of breast cancer.[7 8]

Overall, our findings showed some differences in health-related behaviours between people of different diet groups, thereby highlighting the need to consider such differences when conducting longitudinal analyses in these populations. Future work should also consider possible

differences in other health behaviours between diet groups, such as attendance in colorectal screening. Further study is warranted to understand why people of different diet groups have differential participation in breast screening or prostate cancer testing, HRT use and overall medication use, whether these differences vary by reasons for adhering to each diet group, and whether or how these differences are related to future disease risk.

## Strengths and limitations

This study is the first to simultaneously examine participation in cancer screening or testing, HRT use and medication use in different diet groups. A strength of the study is the large sample size recruited from across different regions in the UK. Additionally, information was collected on a range of factors, which may also be associated with the behaviours of interest, allowing adjustment for these factors. Of potential limitations, recall bias is possible because assessment of the behaviours of interest (ie, breast screening, PSA testing, HRT use and overall medication use) as well as existing medical conditions was based on self-report, although there is no indication that such misclassification bias should differ by diet group. The reasons for which people adhered to each diet group were not recorded, although such reasons may be relevant to the other health behaviours studied. Because of the relatively small number of vegans in our study sample, the role of chance in explaining the findings relating to this diet group, especially subgroup analyses related to medication use, cannot be ruled out. As with most population cohorts, some degree of self-selection and healthy cohort bias may also be present.

## CONCLUSIONS

In this population, we observed differences in breast screening, PSA testing, HRT use and overall medication use between meat eaters, fish eaters, vegetarians and vegans, but no significant differences between diet groups for cervical screening or medication use in people with two or more illnesses or for specific conditions. The reasons for these differences require further investigation. Nonetheless, such differences may be related to or could confound any differences in observed morbidity or mortality from cancer and other diseases between people of different diet groups, and therefore should be considered in future epidemiological studies.

**Acknowledgements** We thank all participants in the EPIC-Oxford cohort for their invaluable contribution.

**Contributors** TYNT, PNA and TJK conceived and designed the research question. TYNT and PNA analysed the data. TYNT wrote the first draft of the manuscript, and PNA, KEB and TJK provided input on data analysis and interpretation of results. All authors revised the manuscript critically for important intellectual content, and read and approved the final manuscript.

**Funding** The work is supported by the UK Medical Research Council MR/M012190/1 and Cancer Research UK 570/A16491 and C9221/A19170. KEB is supported by the Girdlers' New Zealand Health Research Council Fellowship.

**Competing interests** TJK is a member of The Vegan Society. The other authors had no conflicts of interest.

**Ethics approval** The study protocol was approved by a multicentre research ethics committee (Scotland A Research Ethics Committee).

**Provenance and peer review** Not commissioned; externally peer reviewed.

**Data sharing statement** The data access policy for EPIC-Oxford is available via the study website (http://www.epic-oxford.org/data-access-sharing-and-collaboration/).

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
