## [Reviewer comments · BMJ Open]

ARTICLE DETAILS

TITLE (PROVISIONAL)	Cross-sectional analyses of participation in cancer screening and use of hormone replacement therapy and medications in meat eaters and vegetarians: the EPIC-Oxford study.
AUTHORS	Tong, Tammy Yin Ni; Appleby, P; Bradbury, Kathryn; Key, Timothy

VERSION 1 – REVIEW

REVIEWER	Gary E. Fraser Loma Linda University, USA
REVIEW RETURNED	30-Jun-2017

GENERAL COMMENTS	This manuscript addresses questions of compliance with common preventive and therapeutic recommendations by U.K. subjects who also prefer a vegetarian diet. The EPIC-Oxford study is uniquely suited to address this question although 1-2 other very large studies in the U.K. do have moderately large numbers of vegetarians. The methods are generally appropriate and the manuscript is well-written. I have a number of suggestions /questions. 1. The selection of medications that are used for specific conditions (e.g. hypertension, diabetes) was somewhat limited. Is there any information as to what percentage of those with these conditions would be covered by these medications in the U.K? Could apparent shortfalls in treatment be due to the use of alternative medications? There are several other statins, beta blockers, calcium channel blockers ACEI's and ARB's available in most locations. Should be mentioned in the limitations I suspect.2. Were the medical conditions doctor-diagnosed? Any validation data? Again probably a limitation.3. A minor point is that propranolol is mis-spelled in several places.4. Line 132 and related. It does seem that education would be a useful covariate to explore. Are differences by diet mainly because of educational differences? Similarly, it was not clear whether gender was always adjusted for where appropriate.5. Line 136 and on. I was a little confused how this related to the previous paragraph, as it seems that similar issues are covered.6. Line 165. It seems too strong to say that there was no association, particularly for vegans, where the point estimate is quite strong and power is much lower than for other groups.
--

	Can this be expressed somewhat more equivocally? 7. Line 247. One may also wonder whether many vegans and perhaps other vegetarians have an opinion that their lifestyle is protective and they do not need the same focus on preventive screening and “non-natural” therapies. This could be mentioned. 8. The statement in line 259 about the potential confounding by screening is a good point, but needs some modification in that it analyses not adequately adjusted for screening behavior that are problematic. 9. Related to the last point, is there evidence of increased mortality among vegetarians due to delayed screening and more advanced cancers at diagnosis? 10. In Table 4 there are results giving percentages of subjects under appropriate treatment etc. It would be informative to provide a brief comparison to other U.K. data. Was this population unusual in these respects--even the meat-eaters? 11. In Table 5, it is clear that the number of vegans suffering from specific conditions is very small. Drawing conclusions should be very cautions. The prevalence ratios in the final column do not seem to conform to ratios of results in the second column. Is this because results in the 2nd column are not multivariate? Should be clarified.
--	--

REVIEWER	Dr Bernadette Fisher University of Birmingham, UK
REVIEW RETURNED	09-Jul-2017

GENERAL COMMENTS	The rationale for this study is not at all clear in the paper as currently written. I.e. Why and with what purpose in mind is it useful to use four dietary definitions to analyse health/screening behaviours? These are not exactly easy groups to target for health/screening campaigns and the implications of these study findings, e.g. for clinicians, other health practitioners, screening programme promoters etc. needs to be more thoroughly discussed. Otherwise it remains an analysis in search of meaning.
---

REVIEWER	Hanna Skrobanski University of Surrey, United Kingdom
REVIEW RETURNED	19-Sep-2017

GENERAL COMMENTS	This was a nicely written and interesting paper to read. It addressed a research area that very few papers have previously explored and had a large sample size. In the discussion, I found it particularly interesting that it was suggested that non-meat eaters may be reluctant to take medications which are likely to contain animal-derived products. I recommend that this paper is accepted for publication with minor amendments. Here are my suggestions for how the paper could be improved: Abstract - It would be helpful to explain the time points that the follow-up questionnaires were administered. Introduction
---

- In the first sentence of the second paragraph it states: 'the reason for this apparent difference in risk of incident ischaemic heart disease and ischaemic heart disease mortality in vegetarians is unclear'. However, this contradicts the statement in the previous paragraph that no significant difference was found in ischaemic heart disease mortality between diet groups.

- When you describe associations between incidences of specific types of cancer by diet groups being heterogeneous, do you mean for some cancer types the association is non-significant or that meat-eaters have lower risk of incidence for some cancer types? If the second circumstance is the case then explaining this will help to provide a rationale for the suggestion that differences in uptake of cancer screening and testing between diet groups might explain differences in the risk of cancer incidence. By simply providing evidence that vegetarians have lower odds of attending breast and prostate screening, this does not explain why there is a lower risk of overall cancer incidence among this diet group.

- The rationale for determining differences in HRT use by diet groups is not clear. From my understanding HRT is used to alleviate symptoms of the menopause and not to improve or maintain health - it can in fact increase the risk of breast cancer and blood clots.

Methods

- It would be helpful to explain how postal recruitment was targeted at vegetarians, vegans and other people interested in diet and health. The fact that this recruitment method targeted people interested in diet and health may have biased the sample to be over representative of individuals who are more concerned about their health. In the general population, individuals may choose to have a non-meat diet for reasons other than their health (e.g. for the environment or not liking the taste of meat).

- Please state the specific health and lifestyle characteristics that were asked about in the questionnaire and how they were measured.

- It would be useful for the 36 medications and 29 medical conditions that were measured to be specified.

- Please describe how you determined whether women were post-menopausal.

- Please specify how Townsend deprivation was measured.

Result

- It would be helpful for the reader to report the response rate of the full recruitment questionnaire and the retention rates of the two follow up questionnaires.

Discussion

- The discussion could be improved by discussing other known psychosocial determinants of cancer screening and testing (e.g. risk perceptions, fatalistic beliefs, perceived embarrassment and pain, and response efficacy) and HRT and other medication use, and how these determinants may differ by diet group.

- There is no mention of the known association between over diagnosis of cancer and attendance of breast screening and prostate testing.

REVIEWER	Theresa Hastert Wayne State University School of Medicine Karmanos Cancer Institute, USA
REVIEW RETURNED	26-Sep-2017

GENERAL COMMENTS	In this study the authors use the EPIC-Oxford study to examine whether cancer screening, hormone replacement therapy (HRT) use, or use of other medications differed by dietary patterns, comparing vegans, vegetarians, and fish-eaters to meat eaters, and found that compared with meat eaters, fewer vegans or vegetarians reported receiving breast or PSA screening, lower HRT use, and lower use of any medication in the presence of one or no self-reported illnesses. Overall this is a well-written paper that presents some interesting findings. My main comment is that it is not clear what the motivation for this work is. This paper would be strengthened by including a clear statement of why the authors believe it is important to examine these behaviors by diet pattern as well as the authors' hypotheses regarding how they expected diet pattern to be related to these behaviors. Without that, it is difficult to know what to make of these findings. My other comments appear below: Methods: How were vegetarians and vegans targeted? What were the four questions that were used to categorize the diet patterns? A brief sentence or two describing these would be helpful. Why exclude participants missing medication use information from analyses of screening? How was menopausal status determined? Please include additional information on how covariates in the sensitivity analyses were measured/categorized and include basic information about what is included in the Townsend Index. Discussion: The information in the Discussion regarding screening programs and which groups were invited and encouraged to participate in screening. It would be helpful if the authors could clarify whether these programs were in use at the time participants were recruited and data were collected in the 1990s and early 2000s. I suggest rephrasing the statement about "the latter does not appear to involve so much personal choice" on p. 12 lines 237-238 to put less emphasis on the qualitative work which do not seem to be definitive on this issue. For differences in HRT use (Table 3), do the authors think that this is a matter of non-meat eaters having fewer symptoms that would be indications for HRT, or a decision by women experiencing similar symptoms making a decision to forego HRT? More clarity and context for the results and what the authors think is going on and why would greatly strengthen this work.
---

VERSION 1 – AUTHOR RESPONSE

Reviewer: 1

Reviewer Name: Gary E. Fraser

Institution and Country: Loma Linda University, USA

Competing Interests: None declared

This manuscript addresses questions of compliance with common preventive and therapeutic recommendations by U.K. subjects who also prefer a vegetarian diet. The EPIC-Oxford study is uniquely suited to address this question although 1-2 other very large studies in the U.K. do have moderately large numbers of vegetarians. The methods are generally appropriate and the manuscript is well-written. I have a number of suggestions /questions.

1. The selection of medications that are used for specific conditions (e.g. hypertension, diabetes) was somewhat limited. Is there any information as to what percentage of those with these conditions would be covered by these medications in the U.K? Could apparent shortfalls in treatment be due to the use of alternative medications? There are several other statins, beta blockers, calcium channel blockers ACEI's and ARB's available in most locations. Should be mentioned in the limitations I suspect.

R2: We thank the reviewer for his comments. The selection of medications for specific conditions in this paper was based on the 36 named medications on the 10 year follow-up questionnaire, but there is no published information on the percentage of people in the UK with these conditions which would be covered by these medications. We acknowledge that this may be a limitation of the study that we have not covered all medications, and we have added relevant discussion on p.14 lines 281-284: "Since information on medication use in this study was based on a pre-specified list from the follow-up questionnaire, it was not possible to assess the use of alternative therapies or any other named medications, despite their possible contributions to prevalence of overall medication use."

2. Were the medical conditions doctor-diagnosed? Any validation data? Again probably a limitation.

R3: Classification of medical conditions were based on the question "Has your doctor ever told you that you had any of the following?", so the conditions were self-reported. We have now clarified this as a limitation on p.15 lines 317-319:

"Of potential limitations, recall bias is possible because assessment of the behaviours of interest (i.e. breast screening, PSA testing, HRT use and overall medication use) as well as existing medical conditions was based on self-report..."

3. A minor point is that propranolol is mis-spelled in several places.

R4: Thank you for this comment. We have now corrected this.

4. Line 132 and related. It does seem that education would be a useful covariate to explore. Are differences by diet mainly because of educational differences? Similarly, it was not clear whether gender was always adjusted for where appropriate.

R5: Thank you for this suggestion, adding education as a covariate did not make any difference to the estimates. We have now included education level in the reporting of sensitivity analyses in both the methods and results.

“As sensitivity analyses, we repeated the analyses as follows: using data from the 5 year follow-up questionnaire where available; and further adjusting for... and education level (no qualifications, basic secondary e.g. O level, higher secondary e.g. A level, degree, unknown).” p.8-9 lines 159-164

“Results were consistent when we repeated the analyses where possible using data from the 5 year follow-up questionnaire, or when we further adjusted for smoking, alcohol consumption, Townsend deprivation index, and education level (results not shown).” p.10 lines 201 to 204.

Four of the outcomes in this study were gender specific (breast screening, PSA testing, cervical screening, HRT use), and therefore were not adjusted for gender. Analyses for medication use were adjusted for the cross stratification of sex and age at follow-up (p.8 line 147 and footnote of Table 3-5).

5. Line 136 and on. I was a little confused how this related to the previous paragraph, as it seems that similar issues are covered.

R6: We have now edited the text to better reflect the methods on p.8 lines 141-158, which describes the regression model used for all analyses, followed by covariates for analyses of cancer screening or testing and use of HRT, then by analyses of overall medication use, subsequently analyses of medication use stratified by number of self-reported illness or conditions, then finally covariates for the analyses on medication use for specific conditions.

6. Line 165. It seems too strong to say that there was no association, particularly for vegans, where the point estimate is quite strong and power is much lower than for other groups. Can this be expressed somewhat more equivocally?

R7: This has now been edited to read “In men, vegetarians had lower reported uptake of PSA testing (0.82; 0.71, 0.96) than meat eaters, while the difference in uptake appeared lower but did not reach statistical significance in vegans (0.72; 0.50, 1.02), and was not significantly different in fish eaters (0.99; 0.85, 1.07).” p.10 lines 185-188.

7. Line 247. One may also wonder whether many vegans and perhaps other vegetarians have an opinion that their lifestyle is protective and they do not need the same focus on preventive screening and “non-natural” therapies. This could be mentioned.

R8: Thank you for this suggestion, we have now mentioned it on p.13 lines 253-259:

“In studies which assessed attitudes towards cancer screening or testing, common reasons which affect people’s participation in screening include their education level and knowledge of the procedure, recommendation by their doctor, fear of the procedure or the outcome, or their perceived risk of cancer [20–23]. If vegetarians and vegans felt their diets or lifestyles were protective against cancer for example, they might be more likely to forgo cancer screening as a result of lower perceived risk. However, no information was found on whether or how such attitudes may vary by diet group.”

8. The statement in line 259 about the potential confounding by screening is a good point, but needs some modification in that it is analyses not adequately adjusted for screening behavior that are problematic.

R9: Thank you for this suggestion. We have now added to our discussion on p.14 lines 294-296:

“Therefore, future work on assessing breast cancer risk in people of different diet groups should take into account any differences in screening rates between diet groups.”

9. Related to the last point, is there evidence of increased mortality among vegetarians due to delayed screening and more advanced cancers at diagnosis?

R10: We did not find any published evidence on increased mortality among vegetarians due to delayed screening or more advanced cancers, but it will be an interesting future research question.

10. In Table 4 there are results giving percentages of subjects under appropriate treatment etc. It would be informative to provide a brief comparison to other U.K. data. Was this population unusual in these respects--even the meat-eaters?

R11: We have now added discussion of overall medication use comparing EPIC-Oxford participants with the overall UK population, on p.13 lines 271-276:

“The prevalence of medication use in meat eaters (56%) in EPIC-Oxford was slightly higher than the UK average of 43% of men and 50% of women aged 16 or above who reported taking at least one prescribed medicine in the last week [25], confirming the relatively low prevalence of medication use in the vegetarians (39%) and vegans (28%). However, given the differences in age ranges and possible differences in medications accounted for, strict comparisons cannot be made.”

11. In Table 5, it is clear that the number of vegans suffering from specific conditions is very small. Drawing conclusions should be very cautious. The prevalence ratios in the final column do not seem to conform to ratios of results in the second column. Is this because results in the 2nd column are not multivariate? Should be clarified.

R12: The non-conformity between the prevalence ratios in the final column and the ratios of the second column were indeed because the numbers in the second column were crude estimates, and differences were mainly due to differences in age by diet groups. The footnotes of the Table 5 (and other similar tables) have now been edited to reflect this. We have also emphasised in our limitations that “Because of the relatively small number of vegans in our study sample, the role of chance in explaining the findings relating to this diet group, especially subgroup analyses related to medication use, cannot be ruled out.” p.15 line 322-325.

Reviewer: 2

Reviewer Name: Dr Bernadette Fisher

Institution and Country: University of Birmingham, UK

Competing Interests: None declared

Comment: The rationale for this study is not at all clear in the paper as currently written. I.e. Why and with what purpose in mind is it useful to use four dietary definitions to analyse health/screening behaviours? These are not exactly easy groups to target for health/screening campaigns and the implications of these study findings, e.g. for clinicians, other health practitioners, screening programme promoters etc. needs to be more thoroughly discussed. Otherwise it remains an analysis in search of meaning.

R13: We thank the reviewer for her comments. We have now rewritten our introduction to emphasise the rationale of this study, and specifically on p.5 lines 65-74:

“The increasing popularity and interest in vegetarian diets [14] prompts research on the long-term health of vegetarians and vegans. Because health behaviour such as screening or medication use may ultimately influence disease risk, the understanding of any differences in these behaviours by diet group is crucial for the appropriate appraisal of possible differences in disease risk between diet groups. However, current knowledge on this topic is insufficient, because literature on participation in screening and use of medication across people of different diet groups is scarce. Therefore, the aim of this study was to assess some of these relevant health behaviours, including participation in cancer

screening or testing, and use of HRT and other medications among people of different diet groups, in a large population-based cohort in the UK with a high percentage of vegetarians.”

Reviewer: 3

Reviewer Name: Hanna Skrobanski

Institution and Country: University of Surrey, United Kingdom

Competing Interests: None declared

This was a nicely written and interesting paper to read. It addressed a research area that very few papers have previously explored and had a large sample size. In the discussion, I found it particularly interesting that it was suggested that non-meat eaters may be reluctant to take medications which are likely to contain animal-derived products. I recommend that this paper is accepted for publication with minor amendments. Here are my suggestions for how the paper could be improved:

Abstract

- It would be helpful to explain the time points that the follow-up questionnaires were administered.

R14: We thank the reviewer for her comments. This has now been added to the abstract, p.2 line 24.

Introduction

- In the first sentence of the second paragraph it states: ‘the reason for this apparent difference in risk of incident ischaemic heart disease and ischaemic heart disease mortality in vegetarians is unclear’. However, this contradicts the statement in the previous paragraph that no significant difference was found in ischaemic heart disease mortality between diet groups.

R15: We intended to indicate that it is inconsistent that there should be a difference in ischaemic heart disease incidence, but no difference in ischaemic heart disease mortality between diet groups. We have now rewritten the introduction, specifically,
“For cardiovascular diseases, vegetarians in EPIC-Oxford have been observed to have lower ischaemic heart disease risk (hospitalization and death combined) [11], but no significant difference in ischaemic heart disease mortality was observed between diet groups in the same population [12]. The reason for this apparent difference between incidence and mortality is unclear.” p.4 lines 57-61.

Comment: When you describe associations between incidences of specific types of cancer by diet groups being heterogeneous, do you mean for some cancer types the association is non-significant or that meat-eaters have lower risk of incidence for some cancer types? If the second circumstance is the case then explaining this will help to provide a rationale for the suggestion that differences in uptake of cancer screening and testing between diet groups might explain differences in the risk of cancer incidence. By simply providing evidence that vegetarians have lower odds of attending breast and prostate screening, this does not explain why there is a lower risk of overall cancer incidence among this diet group.

R16: We have now rewritten the introduction to improve the overall rationale of the study, and therefore the original wording has been changed.

Comment: The rationale for determining differences in HRT use by diet groups is not clear. From my understanding HRT is used to alleviate symptoms of the menopause and not to improve or maintain health - it can in fact increase the risk of breast cancer and blood clots.

R17: We have now clarified in the introduction on p.4 lines 49-53:

“Because health related behaviours, such as participation in cancer screening [6] or use of hormone replacement therapy (HRT) [7,8], may contribute to the observed rates of cancer, the presence of any differences in these behaviours between diet groups in different populations deserve further investigation.”

The association of HRT and cancer risk was also revisited in the discussion on p.14-15 lines 298-304, specifically, “use of HRT may confound any observed associations between diet group and breast cancer, given that HRT preparations containing oestrogens and progestogens have been shown to increase the risk of breast cancer [7,8]”

Methods

- It would be helpful to explain how postal recruitment was targeted at vegetarians, vegans and other people interested in diet and health. The fact that this recruitment method targeted people interested in diet and health may have biased the sample to be over representative of individuals who are more concerned about their health. In the general population, individuals may choose to have a non-meat diet for reasons other than their health (e.g. for the environment or not liking the taste of meat).

R18: We have clarified in the methods on p.5 lines 84-87:

“The postal recruitment was targeted at vegetarians, vegans, and other people interested in diet and health, by contacting members of The Vegetarian Society, The Vegan Society, and via leaflets enclosed in vegetarian and health food magazines and displayed in health-food shops, and recruited 57,990 participants aged ≥ 20 years.”

The EPIC-Oxford study was established to recruit as many people as possible across a range of diet groups, and therefore the recruitment method targeted vegetarians and vegans not accounting for the reasons they adopted their particular dietary habits. Although the participants in EPIC-Oxford may be more health conscious, some degree of self-selection or healthy cohort bias may be present in any population study. However, we have added as a limitation on p.15 lines 321-322:

“The reasons for which people adhered to each diet group were not recorded, even though such reasons may be relevant to the other health behaviours studied.”

Comment: Please state the specific health and lifestyle characteristics that were asked about in the questionnaire and how they were measured.

R19: We have clarified on p.5-6 lines 87-92 “Altogether, 57,443 participants completed a full recruitment questionnaire which asked about their personal details (including postcode to which a Townsend index of area-level deprivation was assigned [15]), habitual diet and other health and lifestyle characteristics, including personal and family medical history, medication use, socio-economic characteristics, smoking and drinking behaviour, and physical activity levels.”

Comment: It would be useful for the 36 medications and 29 medical conditions that were measured to be specified.

R20: Thank you for this suggestion. We have added this information as Supplementary text 1 and 2.

Comment: Please describe how you determined whether women were post-menopausal.

R21: We have clarified on p.8 lines 138-140 “. For HRT use, we further restricted the analysis to post-menopausal women, determined by including only participants who answered yes to the question ‘Have you been through your menopause?’ on the follow-up questionnaire.”

Comment: Please specify how Townsend deprivation was measured.

R22: We have clarified on p.5-6 lines 88-89 “Altogether, 57,443 participants completed a full recruitment questionnaire which asked about their personal details (including postcode to which a Townsend index of area-level deprivation was assigned [15])...”

Result

- It would be helpful for the reader to report the response rate of the full recruitment questionnaire and the retention rates of the two follow up questionnaires.

R23: The initial response rate of our study could not be estimated due to the study design (the recruitment involved including leaflets in magazines for example). We have added the retention rates of the two follow-up questionnaires on p.9 lines 169-171: “Overall, 57,443 participants in EPIC-Oxford cohort completed a full recruitment questionnaire, of whom 38,043 (66%) completed the 5 year follow-up questionnaire, and 31,695 (55%) completed the 10 year follow-up questionnaire.”

Discussion

- The discussion could be improved by discussing other known psychosocial determinants of cancer screening and testing (e.g. risk perceptions, fatalistic beliefs, perceived embarrassment and pain, and response efficacy) and HRT and other medication use, and how these determinants may differ by diet group.

R24: Although some studies have assessed psychosocial determinants associated with cancer screening or testing, no information was found on how such attitudes could vary by diet group. We have now included in the discussion on p.13 lines 253-259:

“In studies which assessed attitudes towards cancer screening or testing, common reasons which affect people’s participation in screening include their education level and knowledge of the procedure, recommendation by their doctor, fear of the procedure or the outcome, or their perceived risk of cancer [20–23]. If vegetarians and vegans felt their diets or lifestyles were protective against cancer for example, they might be more likely to forgo cancer screening as a result of lower perceived risk. However, no information was found on whether or how such attitudes may vary by diet group.”

Comment: There is no mention of the known association between over diagnosis of cancer and attendance of breast screening and prostate testing.

R25: We have now added on p.14 lines 288-290 “For example, breast cancer screening results in higher incidence but reduced mortality from breast cancer among those who are screened [6]. Prostate cancer testing is also linked to increased incidence in those who are tested [27,28].”

Reviewer: 4

Reviewer Name: Theresa Hastert

Institution and Country: Wayne State University School of Medicine, Karmanos Cancer Institute, USA

Competing Interests: None declared

In this study the authors use the EPIC-Oxford study to examine whether cancer screening, hormone replacement therapy (HRT) use, or use of other medications differed by dietary patterns, comparing vegans, vegetarians, and fish-eaters to meat eaters, and found that compared with meat eaters, fewer vegans or vegetarians reported receiving breast or PSA screening, lower HRT use, and lower use of any medication in the presence of one or no self-reported illnesses.

Overall this is a well-written paper that presents some interesting findings.

My main comment is that it is not clear what the motivation for this work is. This paper would be strengthened by including a clear statement of why the authors believe it is important to examine these behaviors by diet pattern as well as the authors' hypotheses regarding how they expected diet pattern to be related to these behaviors. Without that, it is difficult to know what to make of these findings.

R26: We thank this reviewer for her comments. We have now rewritten the introduction which we believe better reflects the motivation of this work. Please see R13 for further details.

My other comments appear below:

Methods:

How were vegetarians and vegans targeted?

R27: We have clarified in the methods on p.5 lines 84-87: "The postal recruitment was targeted at vegetarians, vegans, and other people interested in diet and health, by contacting members of The Vegetarian Society, The Vegan Society, and via leaflets enclosed in vegetarian and health food magazines and displayed in health-food shops, and recruited 57,990 participants aged ≥ 20 years."

Comment: What were the four questions that were used to categorize the diet patterns? A brief sentence or two describing these would be helpful.

R28: We have clarified on p.6 lines 99-101 "In the recruitment questionnaire and each subsequent follow-up questionnaire, four questions were asked regarding consumption of meat, fish, dairy products, and eggs, in the form of "Do you eat any meat?" or similar for the other three food groups."

Comment: Why exclude participants missing medication use information from analyses of screening?

R29: The reason for this exclusion criterion was that we wanted to ensure an overlapping cohort for analyses i.e. the people assessed for the screening outcomes are a subset of the people used for assessing medication use. We have now explained this on p.7 lines 130-133: "Participants were excluded from all analysis if they did not answer the relevant questions to be assigned to an appropriate diet group (n=28), and in order to ensure that an overlapping population was used for the analyses of all outcomes, they were also excluded if they did not answer the relevant question on medication use (n=407)."

Comment: How was menopausal status determined?

R30: We have clarified on p.8 lines 138-140 "For HRT use, we further restricted the analysis to post-menopausal women, determined by including only participants who answered yes to the question 'Have you been through your menopause?' on the follow-up questionnaire"

Comment: Please include additional information on how covariates in the sensitivity analyses were measured/categorized and include basic information about what is included in the Townsend Index.

R31: We have included additional information on the measurement and categorisation of covariates on p.8-9 lines 159-164:

"As sensitivity analyses, we repeated the analyses as follows: using data from the 5 year follow-up questionnaire where available; and further adjusting for smoking status (never, former, current, unknown), alcohol consumption (<1 g/day, 1-7 g/day, 8-15 g/day, ≥ 16 g/day), Townsend index of area-level deprivation (quartiles and unknown), and education level (no qualifications, basic secondary e.g. O level, higher secondary e.g. A level, degree, unknown)."

Calculation of Townsend Index is usually based on unemployment, non-car ownership, non-home ownership, and household overcrowding, but specifically in EPIC-Oxford, existing area-level values were assigned based on participants' postcodes. We believe it is beyond the scope of this paper to include detailed information on the Townsend score, but we have clarified that on p.5 lines 87-89 "Altogether, 57,443 participants completed a full recruitment questionnaire which asked about their personal details (including postcode to which a Townsend index of area-level deprivation was assigned [15])."

Discussion:

The information in the Discussion regarding screening programs and which groups were invited and encouraged to participate in screening. It would be helpful if the authors could clarify whether these programs were in use at the time participants were recruited and data were collected in the 1990s and early 2000s.

R32: Both breast screening and cervical screening programmes were introduced in 1988 in the UK, and therefore both programmes had been in place for a period of time at the 5 (2000-2003) and 10 (2007) year follow up. We have now added to the relevant section on p.12 lines 249-251 "In the UK since 1988 [6,16], all women aged 50 to 70 are invited to attend breast cancer screening clinics [17] and all women aged 25 to 64 are invited for cervical screening [18] at regular intervals."

Comment: I suggest rephrasing the statement about "the latter does not appear to involve so much personal choice" on p. 12 lines 237-238 to put less emphasis on the qualitative work which do not seem to be definitive on this issue.

R33: Thank you for this suggestion. We have now rephrased the section on p.13 lines 260-264: "In a small focus group study in Scotland which asked participants about their attitudes towards cancer screening (n=31 for cervical screening, n=10 for breast screening), the study participants reported that they felt pressure from health care professionals, family and friends to attend cervical screening but not breast screening, and that they also considered cervical screening to be normative routine behaviour [24]."

Comment: For differences in HRT use (Table 3), do the authors think that this is a matter of non-meat eaters having fewer symptoms that would be indications for HRT, or a decision by women experiencing similar symptoms making a decision to forego HRT? More clarity and context for the results and what the authors think is going on and why would greatly strengthen this work.

R34: Thank you for this comment. There is no literature or evidence to suggest which scenario is more likely. We have now added to our discussion on p.14 lines 298-330: "Similarly, it is not clear why there was differential use of HRT in the four diet groups, for example whether it was because non-meat eaters were less likely to have symptoms, or because they were less likely to seek treatment when symptoms appear."

VERSION 2 – REVIEW

REVIEWER	Dr Bernadette Fisher University of Birmingham , UK
REVIEW RETURNED	15-Nov-2017

GENERAL COMMENTS	The reason I have not recommended this paper for publication is that, try as I might, I cannot see the value in examining whether or not individuals with different dietary behaviours also have different screening behaviours, unless this is also accompanied by some insights into a) their reasons for choosing their particular dietary habits, i.e. health related or not; b) their beliefs and attitudes towards health and specifically screening. Their dietary choices may not reflect any inherent differences in health beliefs and attitudes. The authors also acknowledge that this data is needed to make more sense of their findings and, my view is that without it, useful inferences cannot be confidently made. Even explicitly targeting 'vegetarian's or 'vegans' in publicity materials promoting preventative health screening, would require understanding whether the differences in health screening behaviour, noted among these 'diet' subgroups is or is not related to their wider health beliefs and attitudes. A final point concerns the choice of diet to define the sub groups examined. As a highly complex composite of many behaviours and choices, diet has, to date, only been weakly and quite inconsistently associated with the development of breast, cervical and prostate cancers. Given the much stronger relationships found with physical activity, alcohol and obesity, it might have been more relevant to examine screening behaviours within these sub groups, as then educational materials, focusing on both the value of screening for detection and risks/benefits associated with these behaviours, could quite possibly have been developed for testing. In summary, I think you need to do the belief/attitudinal work to overlay these findings, which could then have some value in developing more targeted public health messages.
--

REVIEWER	Hanna Skrobanski University of Surrey, United Kingdom
REVIEW RETURNED	07-Nov-2017

GENERAL COMMENTS	Having reviewed the first version of the manuscript, I am happy that all my previous comments have been addressed. The manuscript was an enjoyable and interesting read. The study addresses a novel research area during a time when vegetarianism is becoming increasingly popular. Therefore, exploring differences in health behaviours between diet groups warrants attention. My only suggestions for revisions are: 1) To explain why participants were only asked whether they had ever had cervical screening by the smear test in the 5 year follow-up questionnaire.2) To explain why the question on medication use on the 5 year questionnaire was shorter than the 10 year questionnaire.3) To acknowledge in the discussion that differences in colorectal screening attendance between diet groups also warrants attention.
---

REVIEWER	Theresa Hastert Wayne State University, Karmanos Cancer Institute, USA
REVIEW RETURNED	08-Nov-2017

GENERAL COMMENTS	Thank you for addressing my previous comments.
--

VERSION 2 – AUTHOR RESPONSE

Reviewer: 2

Reviewer Name: Dr Bernadette Fisher

Institution and Country: University of Birmingham , UK Competing Interests: None declared

Comment: The reason I have not recommended this paper for publication is that, try as I might, I cannot see the value in examining whether or not individuals with different dietary behaviours also have different screening behaviours, unless this is also accompanied by some insights into a) their reasons for choosing their particular dietary habits, i.e. health related or not; b) their beliefs and attitudes towards health and specifically screening. Their dietary choices may not reflect any inherent differences in health beliefs and attitudes. The authors also acknowledge that this data is needed to make more sense of their findings and, my view is that without it, useful inferences cannot be confidently made. Even explicitly targeting 'vegetarian's or 'vegans' in publicity materials promoting preventative health screening, would require understanding whether the differences in health screening behaviour, noted among these 'diet' subgroups is or is not related to their wider health beliefs and attitudes. A final point concerns the choice of diet to define the sub groups examined. As a highly complex composite of many behaviours and choices, diet has, to date, only been weakly and quite inconsistently associated with the development of breast, cervical and prostate cancers. Given the much stronger relationships found with physical activity, alcohol and obesity, it might have been more relevant to examine screening behaviours within these sub groups, as then educational materials, focusing on both the value of screening for detection and risks/benefits associated with these behaviours, could quite possibly have been developed for testing. In summary, I think you need to do the belief/attitudinal work to overlay these findings, which could then have some value in developing more targeted public health messages.

R2: We thank the reviewer for her comments. The purpose of this work was not to make an immediate public health impact by promoting screening in any particular diet group, but was rather to provide a description of observed differences in health behaviours in people of different diet groups, which may subsequently influence their disease risk. As the reviewer mentioned, diet has been weakly and inconsistently associated with various cancers, and one contributing factor to this inconsistency in evidence may be related to residual confounding in different studies. Therefore, we feel it is important to document the observed differences of some possible factors that may confound the association between diet group and disease risk, regardless of the reasons for adhering to any particular diet group. For example, as we mentioned in lines 292-299 "using breast cancer as an example, given the lower rates of breast cancer screening among non-meat eating women both in EPIC-Oxford and in the Swedish Malmö Diet and Cancer Study [9], it is possible that the observed incidence of breast cancer in these diet groups underestimates the true incidence owing to detection bias, but that ultimately these women would be expected to have a somewhat higher mortality from breast cancer. Therefore, future work on assessing breast cancer risk in people of different diet groups should take into account any differences in screening rates between diet groups."

Overall, we believe that the differences in health behaviours by diet groups observed in our study deserve to be reported and documented even in the absence of reason for adhering to the diet (this information was not collected in our study), as understanding of such differences is crucial for better interpreting the true association between diet and health. We have previously acknowledged that “The reasons for which people adhered to each diet group were not recorded, although such reasons may be relevant to the other health behaviours studied.” (lines 325-326). We have now additionally mentioned that “Further study is warranted to understand why people of different diet groups have differential participation in breast screening or prostate cancer testing, HRT use, and overall medication use, whether these differences vary by reasons for adhering to each diet group, and whether or how these differences are related to future disease risk.” (lines 311-314).

Reviewer: 3

Reviewer Name: Hanna Skrobanski

Institution and Country: University of Surrey, United Kingdom Competing Interests: None declared

Having reviewed the first version of the manuscript, I am happy that all my previous comments have been addressed. The manuscript was an enjoyable and interesting read. The study addresses a novel research area during a time when vegetarianism is becoming increasingly popular. Therefore, exploring differences in health behaviours between diet groups warrants attention.

My only suggestions for revisions are:

1) To explain why participants were only asked whether they had ever had cervical screening by the smear test in the 5 year follow-up questionnaire.

R3: We thank the reviewer for her comments. We have now added to the manuscript “Due to the changing research focus over the course of data collection, slight variations existed between questions asked on the 5 and 10 year follow-up questionnaires.” (lines 96-98)

2) To explain why the question on medication use on the 5 year questionnaire was shorter than the 10 year questionnaire.

R4: The reason for this difference was the same as above (R3).

3) To acknowledge in the discussion that differences in colorectal screening attendance between diet groups also warrants attention.

R5: We have now added to the manuscript “Future work should also consider possible differences in other health behaviours between diet groups, such as attendance of colorectal screening.” (lines 309-311)

Reviewer: 4

Reviewer Name: Theresa Hastert

Institution and Country: Wayne State University, Karmanos Cancer Institute, USA Competing Interests: None declared

Thank you for addressing my previous comments.